# Immobilization of Cellulolytic Enzymes in Accurel® MP1000

**Julia R. S. Baruque, Adriano Carniel** [ID]**, Júlio C. S. Sales, Bernardo D. Ribeiro** [ID]**, Rodrigo P. do Nascimento** [ID]
**and Ivaldo Itabaiana, Jr. *** [ID]

Department of Biochemical Engineering, School of Chemistry, Federal University of Rio de Janeiro,
Rio de Janeiro 21941-919, Brazil; jubabaruque@hotmail.com (J.R.S.B.); adrianocarniel@gmail.com (A.C.);
juliocesarsales@gmail.com (J.C.S.S.); bernardo@eq.ufrj.br (B.D.R.); rodrigopires@eq.ufrj.br (R.P.d.N.)
* Correspondence: ivaldo@eq.ufrj.br; Tel.: +55-2139-387-580

**Abstract:** Cellulases are a class of enzymes of great industrial interest that present several strategic applications. However, the high cost of enzyme production, coupled with the instabilities and complexities of proteins required for hydrolytic processes, still limits their use in several protocols. Therefore, enzyme immobilization may be an essential tool to overcome these issues. The present work aimed to evaluate the immobilization of cellulolytic enzymes of the commercial enzyme cocktail *Celluclast*® 1.5 L in comparison to the cellulolytic enzyme cocktail produced from the wild strain *Trichoderma harzianum* I14-12 in Accurel® MP1000. Among the variables studied were temperature at 40 °C, ionic strength of 50 mM, and 72 h of immobilization, with 15 mg·mL$^{-1}$ of proteins generated biocatalysts with high immobilization efficiencies (87% for ACC-*Celluclast* biocatalyst and 95% for ACC-*Th*I1412 biocatalyst), high retention of activity, and specific activities in the support for CMCase (DNS method), FPase (filter paper method) and β-glucosidase (*p*-nitrophenyl-β-D-glucopyranoside method). Presenting a lower protein concentration (0.32 mg·mL$^{-1}$) than the commercial *Celluclast*® 1.5 L preparation (45 mg·mL$^{-1}$), the ACC-*Th*I1412-derived immobilized biocatalyst showed thermal stability at temperatures higher than 60 °C, maintaining more than 90% of the residual activities of FPase, CMCase, and β-glucosidase. In contrast, the commercial-free enzyme presented a maximum catalytic activity at only 40 °C. Moreover, the difference in molecular weight between the component enzymes of the extract was responsible for different hydrophobic and lodging interactions of proteins on the support, generating a robust and competitive biocatalyst.

**Keywords:** cellulases; cellulase immobilization; Accurel® MP1000; *Celluclast*® 1.5 L; *Trichoderma harzianum*



## 1. Introduction

With the world population's increased life expectancy and higher demands for food and energy, biotechnology has turned its attention to developing more sustainable processes, both in academia and in industry. In this context, residual lignocellulosic biomass has emerged as an alternative natural resource since it is a cheap, renewable, and abundant raw material to obtain several high-value-added compounds [1]. As of 2017, the generated residual biomass has reached a value of more than 50 billion tons per year. Despite several technologies already being implemented for the reuse of such wastes, less than 10% in residual mass is reinserted in the production chain, where the study of new protocols of valorization is still a challenge for science [2].

Fermentative processes using residual lignocellulosic biomass as a substrate to obtain enzymes with high industrial applicability have been a very important tool in the valorization of these commodities [3], where competitive enzyme cocktails with lower costs and outlined catalytic properties can be obtained according to the choice of the microorganism used [4]. Several species of filamentous fungi present the ability to produce and secrete a wide range of enzymes, thus making it possible to obtain enzyme extracts with a vast complexity of proteins according to fermentative conditions. Besides high productivity, the capacity for growth and adaptation in simple culture media makes these microorganisms

very attractive to enzyme technology. The global enzyme market is estimated to reach the USD 950 million mark by 2024, when Brazil, France, and the United States will present a prominent position [5].

Among the microbial proteins of great industrial importance, cellulolytic enzymes, such as cellulases and hemicellulases, play an essential role in valorizing residual biomass, thereby having applications in several industrial sectors, such as textile, food, pulp and paper, and biofuels [6,7]. Cellulases are classified as a complex of enzymes responsible for the hydrolysis of the cellulose fraction by acting directly on β-1,4-type glycosidic bonds, leading to the obtainment of glucose, cellobiose, and several other bioproducts. The complete hydrolysis of cellulose to glucose requires the synergistic action of at least three classes of cellulases: endoglucanases (EG; EC 3.2.1.4), exoglucanases (also known as cellobiohydrolases, CBH; EC 3.2.1.91), and β-glucosidases (BG; EC 3.2.1.21) [8]. However, the application of cellulases on an industrial scale still presents some challenges: Besides the low catalytic stability of most enzymes of the cellulolytic complex, the commercial production of these enzyme cocktails requires extensive microbial genetic modification and production under controlled and complex fermentative medium and separation steps [9]. The application of these complexes in their free form generates cocktails that are more susceptible to harmful actions in the reaction medium, thus reducing catalytic efficiency and hindering the utilized biocatalyst's recovery. In this context, enzyme immobilization processes have been a significant step in the design of a bioprocess that achieves better yields and facilitates the separation of proteins from the complex fermentative medium used [10]. Besides associating enzymes with a solid or semi-solid phase, different immobilization techniques can increase the stability of proteins and promote the retention and improvement of their activity and recyclability. Immobilized enzymes can also reduce, in the long term, the operational costs and increase the range of reactors to be used in several processes [11]. The success of a robust biocatalyst depends on several factors, such as the immobilization method and the characteristics of the support used, in addition to the quantitative ratio and affinity between the support and the enzyme, among others [12]. Among the various techniques of enzyme immobilization reported, the adsorption method has gained great prominence in recent years due to it being a simple and relatively cheap technique that does not require the preliminary steps of support functionalization.

Moreover, enzymes can be adsorbed to a support surface in distorted conformations through adsorption techniques, thus increasing their stability and catalytic efficiency [13]. Our research group has relevant experience in enzyme immobilization on hydrophobic supports such as Accurel® MP1000, a porous hydrophobic propylene polymer with a large surface area for adsorption [13,14]. Due to these characteristics, it has been used in many immobilization studies for lipase, laccase, and esterase immobilization [15–19]. However, there is no work in the literature on the immobilization of cellulases on this support. In this context, the present work evaluated the effect of immobilization of the commercial cocktail of cellulases *Celluclast*® 1.5 L in Accurel® MP1000, as well as the application of the best conditions of the process in the study of the immobilization of an enzymatic extract containing cellulases produced from a wild strain of *Trichoderma harzianum* I14-12 by applying sugar cane straw as a carbon source and millet as a nitrogen source.

## 2. Results and Discussion

### 2.1. Immobilization of Commercial Celluclast® 1.5 L in Accurel® MP1000

We started this work by determining the number of proteins and the activities of cellulolytic enzymes present in both the enzymatic cocktail *Celluclast*® 1.5 L and in the enzymatic extract obtained from the *Trichoderma harzianum* I14-12 (*Th* I14-12) strain as the initial parameters for the immobilization steps in Accurel® MP1000, which results are summarized in Table 1.

**Table 1.** Enzymatic activities and amount of protein present in the commercial enzyme cocktail *Celluclast*® 1.5 L and in the home-made enzymatic extract from the strain *Trichoderma harzianum* I14-12 (*Th* I14-12).

| Enzymatic Extract | Enzyme Activity (U·L$^{-1}$) | | | Amount of Protein (mg·mL$^{-1}$) |
|---|---|---|---|---|
| | CMCase | FPase | β-glucosidase | |
| *Celluclast*® 1.5 L | 2118 | 833 | 31,130 | 45 |
| *Th* I14-12 | 2144 | 819 | 21,560 | 0.32 |

As observed, in the enzymatic extract *Th* I14-12, both similar distribution and activities of the enzymes CMCase, FPase, and β-glucosidase could be found when compared to *Celluclast*® 1.5 L, even at 140× lower protein concentrations (45 mg·mL$^{-1}$ for *Celluclast*® 1.5 L versus 0.32 mg of protein per L for the extract Th I14-12). These proteins of the cellulolytic complex are crucial to perform the total hydrolysis of the cellulose fraction of the lignocellulosic biomass [20]. These data are important since *Celluclast*® 1.5 L is an enzymatic preparation of Novozymes® rich in cellulases from a genetically modified strain of *Trichoderma reezei*, which is a known producer of CMCase, FPase, and β-glycosidase [20,21]. However, through several genetic engineering maneuvers, besides producing modified and modeled enzymes, the engineered microorganism can secrete high concentrations of proteins in the medium, and probably for this reason, the amount of protein in the commercial preparation is much higher than the home-made extract obtained from the wild-type *Trichoderma harzianum* I14-12 strain [20,22]. However, our preparation was obtained under simple fermentation conditions and using residual renewable sources of carbon and nitrogen, making it a cheaper, more ecological, and interesting alternative for future industrial applications, which is also proven to be composed of highly active proteins.

In this work, because there was a similar composition in enzymatic activities of the cellulolytic complex in both extracts, and with the aim of performing a comparative study of the immobilization of the commercial and home-made cellulolytic extracts on the Accurel® MP1000 support, the best immobilization conditions on the support were outlined for the commercial preparation and then applied to the extract *Th* I14-12. For this purpose, we started a *Celluclast*® 1.5 L concentration of 45 mg·mL$^{-1}$ in the immobilization media. To determine the best immobilization conditions, two different initial temperatures (20 and 40 °C) and two different ionic strengths (25 and 50 mM of sodium citrate buffer) were investigated for 24 h. The results of the immobilization yield and recovered activities are shown in Table 2.

**Table 2.** Yields and recovered activities in support during the immobilization of *Celluclast*® 1.5 L in Accurel® MP1000. Immobilization conditions: 45 mg·mL$^{-1}$ of enzymatic extract in 50 mL of 25 mM or 50 mM of citrate buffer at 20 °C or 40 °C and 100 rpm.

| Immobilization Conditions | Immobilization Yield (%) | Recovered Activity | | |
|---|---|---|---|---|
| | | CMCase (%) | FPase (%) | β-glucosidase (%) |
| 20 °C, 25 Mm | 10 | 0.16 | 0.5 | 0.015 |
| 20 °C, 50 mM | 24 | 0.92 | 0.6 | 0.02 |
| 40 °C, 25 mM | 32 | 1.28 | 0.81 | 0.03 |
| 40 °C 50 mM | 40 | 1.45 | 1.04 | 0.03 |

It was observed that applying a protein concentration of 45 mg·mL$^{-1}$, the highest temperature (40 °C), and the highest ionic strength (50 mM) resulted in the highest immobilization efficiency, in terms of quantity of proteins in the supernatant, when evaluated during the initial 24 h of the process (kinetic curves shown in Figure S1 of Supplementary Materials).

In an immobilization process via adsorption, the success of the technique used depends on several factors, including the hydrophobicity of the support, the pH of the medium,

and the ionic strength and temperature, so that the enzyme is in its most non-ionized form as much as possible, thus maintaining solubility [23]. Temperature is a factor of great importance since, besides contributing to the solubility of a protein in an immobilization system, it contributes to an enzyme reaching various conformations that may be more compatible with the support, thereby contributing to the better adsorption of the protein on the support [24]. Moreover, temperature can also cause greater movement of the protein content in the immobilization system, thus contributing to better adsorption. It is also important to note that the Accurel® MP1000 support applied in this work is porous. Several authors have reported that in these materials, the increase in temperature may favor the migration of proteins into the pores since this phenomenon is endothermic [24–28]. Ionic strength is directly linked to the solvation of a protein in an immobilization medium, where at low concentrations, the protein is found with a small solvation layer, thus generating larger contact surfaces between the enzyme and the support [25]. However, when the ionic strength becomes higher, more significant amounts of ions are added to the immobilization solution, causing the protein macromolecule to be less solvated and reducing its solubility in the medium, which means it may also interact with the support [25,29]. Therefore, in immobilization studies of extracts containing different enzymes, as in our work, it is of great importance to study this variable in the system.

In Table 1, it is also possible to observe that during the 24 h of immobilization, relatively low values of recovered activity are obtained for CMCase (1.45%), FPase (1.04%), and β-glucosidase (0.03%). These values can be better understood by observing the curves of enzymatic activity of the supernatant over time (Figure 1). Although these curves show a tendency to decrease during the 24 h at 50 mM and 40 °C, it is still possible to detect high residual enzymatic activity, demonstrating that more time of immobilization would be necessary to observe the stabilization of enzymatic activities.

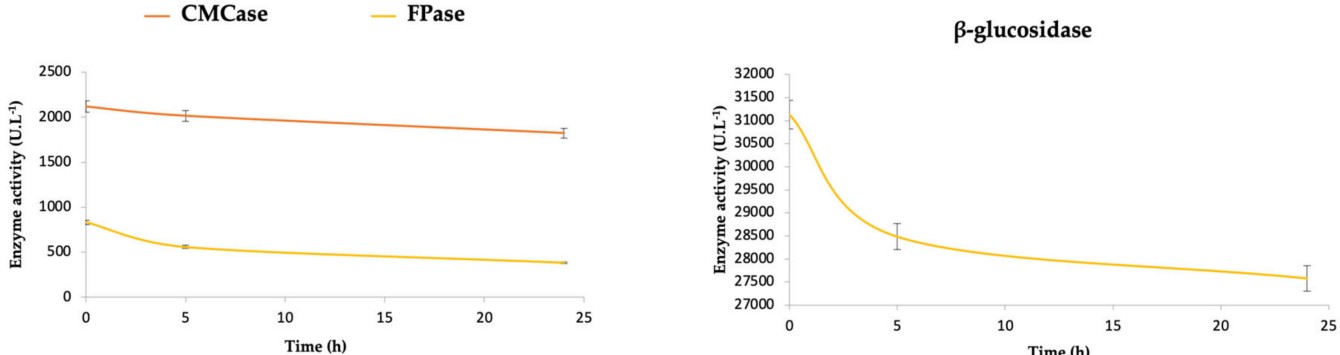

**Figure 1.** Enzymatic activity curves showing decreases over time during the immobilization of *Celluclast*® 1.5 L preparation. Immobilization conditions: 45 mg·mL$^{-1}$ of enzymatic extract in 50 mL of 50 mM citrate buffer at 40 °C and 100 rpm.

Therefore, in order to verify the influence of time on the immobilization process, the adsorption kinetics of *Celluclast*® 1.5 L in Accurel® MP1000 was performed at various times up to 96 h. The profile of residual protein concentration in the supernatant is shown in Figure 2.

As observed, at 48 h of immobilization, more than 50% of all the proteins present in the enzymatic extract has been adsorbed on the support, according to the applied method. In order to understand the characteristics of the enzymes immobilized on the support, the due immobilization efficiencies and recovered activities were also calculated over time, which results are shown in Table 3.

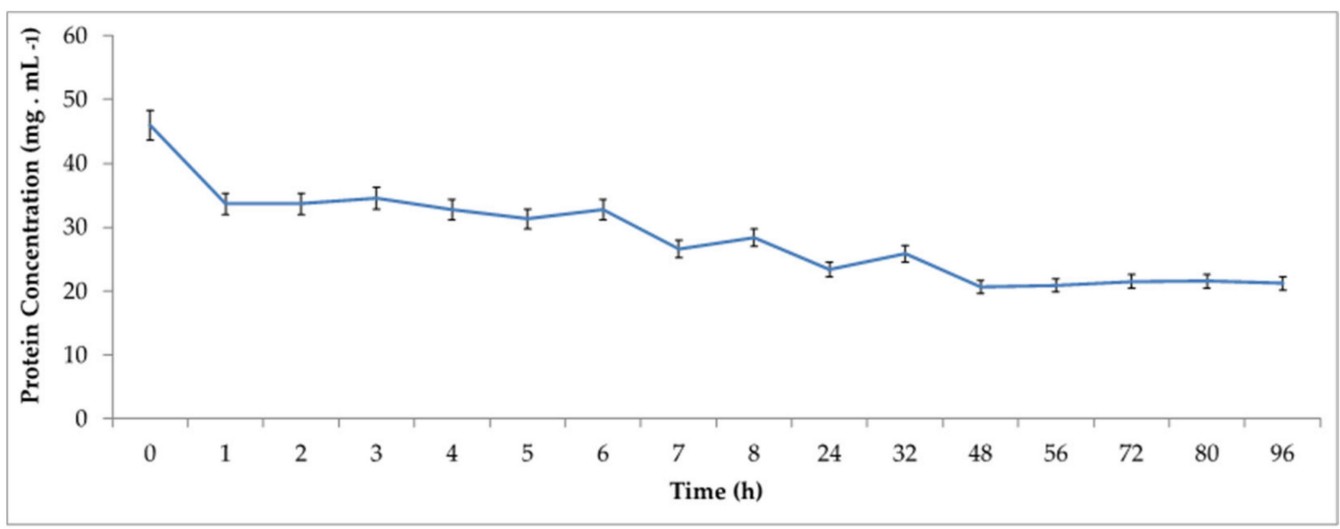

**Figure 2.** Adsorption kinetics of the supernatant during the immobilization process of *Celluclast*® 1.5 L cocktail in Accurel® MP1000. Immobilization conditions: 45 mg·mL$^{-1}$ of enzymatic extract in 50 mL of 50 mM citrate buffer at 40 °C and 100 rpm.

**Table 3.** Yields during the process and activities recovered on the support after 96 h of immobilization at 40 °C and 50 mM of ionic strength. Immobilization conditions: 45 mg·mL$^{-1}$ of enzymatic extract in 50 mL of 50 mM citrate buffer at 40 °C and 100 rpm.

| Time (h) | Immobilization Yield (%) | Recovered Activity (%) | | |
|---|---|---|---|---|
| | | CMCase | FPase | β-glucosidase |
| 24 | 49 | 3.9 | 0.89 | 0.09 |
| 48 | 56 | 3.78 | 0.88 | 0.086 |
| 72 | 55 | 4 | 1 | 1.2 |
| 96 | 55 | 3.4 | 0.97 | 0.083 |

Compared to the 24 h immobilization time, at 72 h, 55% of immobilization yield was achieved, with the percentage of proteins adsorbed on the support stabilized at higher times. However, although with slightly higher values than those found at 24 h of immobilization, the recovered activities of CMCase (4%), FPase (1%), and β-glucosidase (1.2%) are still low. The enzyme activity profiles over time present the same profile (Figure S2 of Supplementary Materials). The commercial *Celluclast*® 1.5 L cocktail is known to offer a different amount among these three enzymes, where it is more enriched with endoglucanases, such as CMCase and FPase, which are enzymes vital for the access of cellulose fibers that are highly recalcitrant, crystalline, and branched biopolymers [21]. Therefore, in order to achieve successful hydrolysis and to maximize the obtainment of oligosaccharides and cellobiose, endoglucanases are key proteins in this process, in addition to the various accessory proteins already reported [30,31]. In this sense, β-glucosidase is, in fact, activated in high concentrations of cellobiose, generating glucose as the final product. However, even in lower protein concentrations, as it is a highly specific enzyme, its activity is higher compared to endoglucanases present in the extract.

Another important factor to consider at this point in the experiments is the initial protein concentration fed to the support. Because at this point of the experiments, a protein concentration of 45 mg·mL$^{-1}$ was applied in order to verify if there was a possible saturation point of the support, and immobilizations of the enzyme extract were performed at lower concentrations in 48 h of incubation. The results are shown in Table 4.

**Table 4.** Immobilization efficiencies and recovered activities of *Celluclast*[®] 1.5 L immobilization on Accurel[®] MP1000 according to the initial protein loading. Immobilization conditions: 5–45 mg·mL$^{-1}$ of enzymatic extract in 50 mL of 50 mM citrate buffer at 40 °C and 100 rpm for 48 h.

| Protein Loading (mg·mL$^{-1}$) | Immobilization Yield (%) | Recovered Activity (%) | | |
|---|---|---|---|---|
| | | CMCase | FPase | β-glucosidase |
| 45 | 55 | 4 | 1 | 1.2 |
| 30 | 62 | 18.6 | 10.4 | 6.1 |
| 15 | 87 | 42.5 | 34.6 | 16.5 |
| 10 | 91 | 10.5 | 18.3 | 9.3 |
| 5 | 100 | 11.2 | 12.4 | 8.3 |

As observed, the immobilization efficiencies showed an increase proportional to the reduction in the amount of protein offered to the support, demonstrating that, in fact, the support could present a protein loading capacity lower than 45 mg·mL$^{-1}$, using the initial amount of our experiments with *Celluclast*[®] 1.5 L. From 15 mg·mL$^{-1}$ onward, immobilization efficiencies close to 100% were found, demonstrating that practically all the proteins present in the enzymatic extract were associated with the support Accurel[®] MP1000. In an immobilization process via adsorption, not only the amount of adsorbed proteins is essential, but so is the last activity expressed in the biocatalyst [32]. Therefore, an excess of adsorbed proteins on the support can generate several layers of enzyme deposition, where part of the proteins may not be completely accessible to interact with the substrate, resulting in low values of recovered activity [32,33]. Thus, the study of the optimal ratio between enzyme and support is of great importance when expressing the maximum catalytic activity of an immobilized biocatalyst [34]. Moreover, the characteristics of the support used, such as surface area, size, porosity, and hydrophobicity, can influence the immobilization protocols and are of great importance in predicting the interactions to be performed with enzymes [33,35].

Aiming to contextualize the characteristics of the support and the enzyme with the results found, SEM analysis of Accurel[®] MP1000 was performed before and after *Celluclast*[®] 1.5 L immobilization at 15 mg·mL$^{-1}$ (Figure 3).

According to the microscopic results, it is possible to observe that Accurel[®] MP1000 is an amorphous and multiporous resin, which is regularly distributed throughout the extension of the polymer (Figure 3A,C,E). According to all the magnifications performed, it could also be better observed that the presence of several filled pores after the immobilization process (Figure 3B,D,F) demonstrates a possible lodging effect of the enzymes inside the pores. Our research group demonstrated in previous work that the surface hydrophobicity of Accurel[®] MP1000, in addition to the distribution and pore size, was important for multiple adsorption interactions and accommodation of lipases, thus generating a more active biocatalyst and a higher thermal stability, with a low percentage of desorption or leaching into the reaction medium [13,14].

In order for these variables to be actually evaluated in the immobilization process, Accurel[®] MP1000 was submitted to analysis using thermogravimetry (TGA), which results are demonstrated in Figure 4.

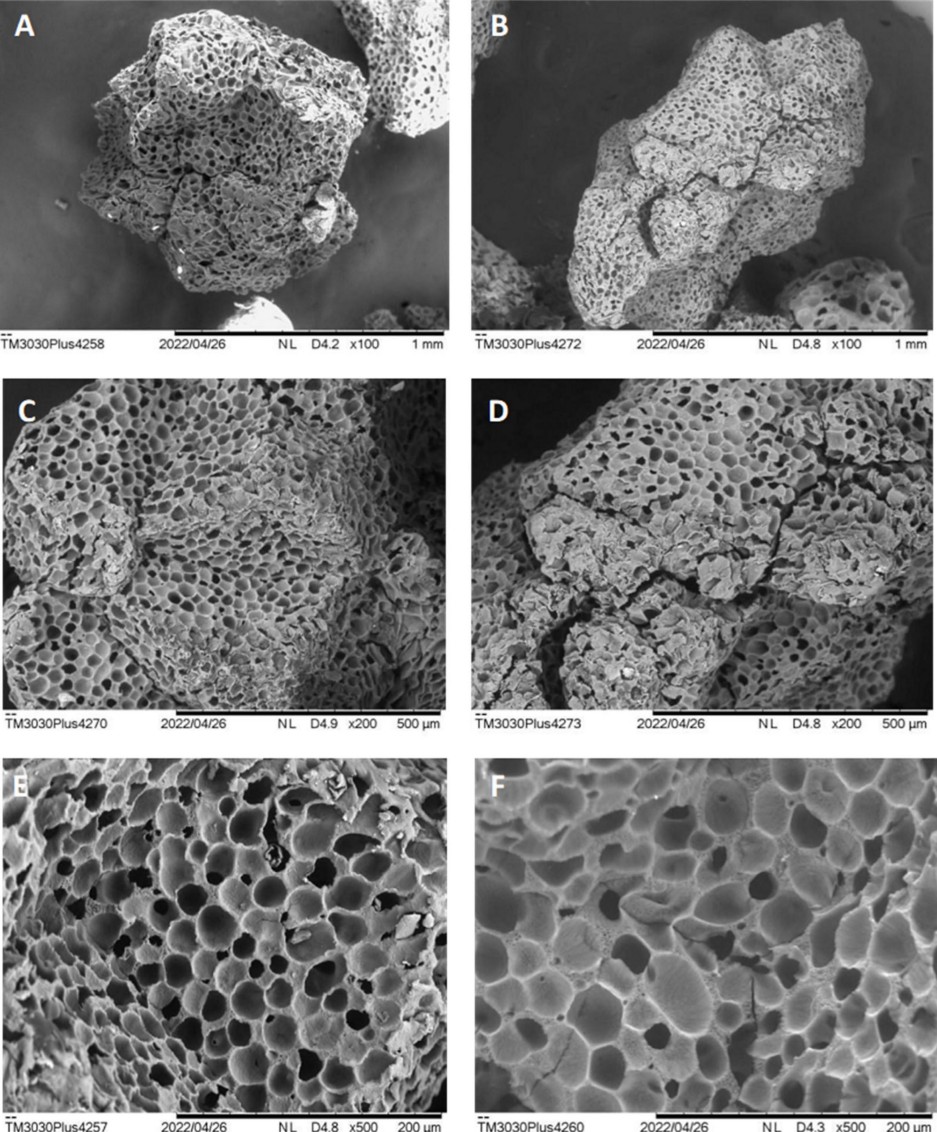

**Figure 3.** SEM analysis of Accurel® MP 1000 before and after 48 h of enzymatic immobilization of Cll. Images (**A**,**C**,**E**) show the support before the process at the magnifications of 100×, 200×, and 500×, respectively. Images (**B**,**D**,**F**) show the support after this procedure at the magnifications of 100×, 200×, and 500×, respectively.

Moreover, the determination of surface area, porous volume, and porous particle size was also performed using the BET method, as well as via hydrophobicity calculation using the Bengal rose method. The results are shown in Table 5.

**Table 5.** Results of superficial area, porous volume, porous particle size, and hydrophobicity of Accurel® MP 1000 as calculated using the Bengal rose method.

| Hydrophobicity ($\mu g \cdot g^{-1}$) | BET Surface Area ($m^2 \cdot g^{-1}$) | Pore Volume ($cm^3 \cdot g^{-1}$) | Pore Size (Å) | Average Particle Size (Å) |
|---|---|---|---|---|
| 22.7 | 5.6 | 0.025 | 184.49 | 10.716 |

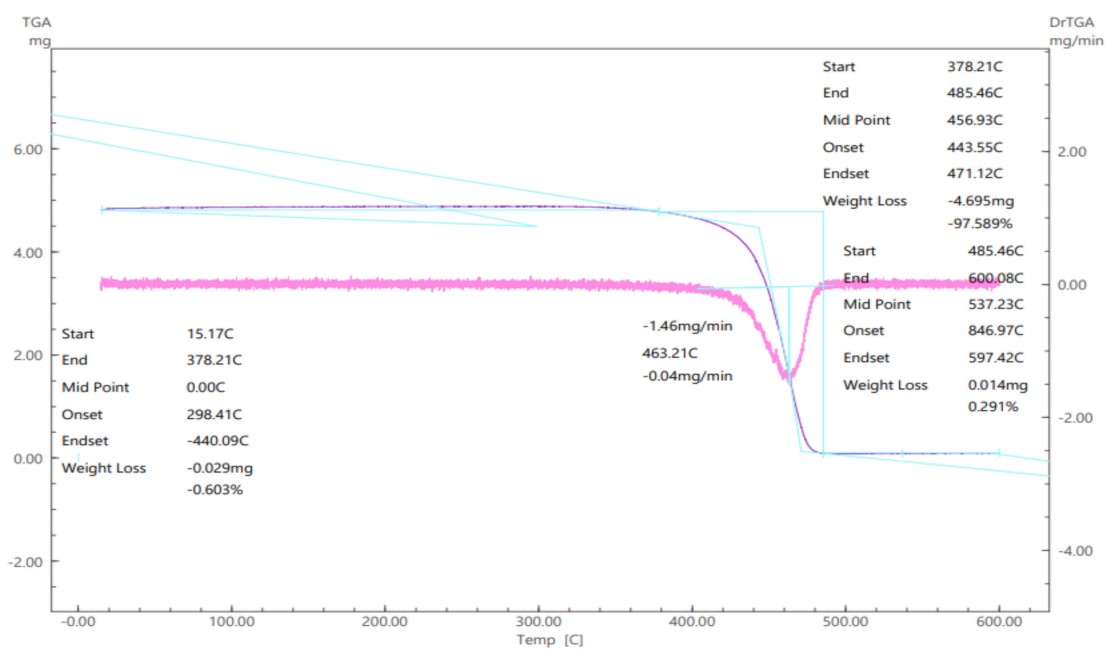

**Figure 4.** Thermogravimetric analysis profile of Accurel® MP 1000.

According to the data obtained, Accurel® MP1000 presents a hydrophobicity of 22.7 μg.g$^{-1}$, which characterizes it as a highly hydrophobic support, which is favorable for the realization of hydrophobic adsorption methodologies since some enzymes, when interacting with hydrophobic supports, can be associated with the support with distorted or more stable conformations, where the catalytic site is more exposed, or can be in a conformation more favorable to recognition by the substrate [35,36].

Besides the hydrophobicity of the support, the surface area, pore size, volume and distribution, and the size of the support granules can influence the immobilization process [34,37]. The size of the pores, for example, needs to be compatible with the size and volume of the enzyme so that the enzyme is correctly accommodated and remains stable. Otherwise, there is a risk of leaching and loss of activity [38,39].

The surface area results obtained via the BET method in this work (5.6 m$^2$.g$^{-1}$) are lower than those found by Singh and collaborators (30 m$^2$.g$^{-1}$) in 2011 [4], which may be due to the batch used, the manufacturer, or even a lower sensitivity of the applied method, given the time of execution of the experiments by the authors. However, even though the value is lower, our results still demonstrate a high surface area of the support, which, combined with the high hydrophobicity found, makes it suitable for enzyme immobilization via surface adsorption methodology. According to Zdravkov and collaborators in 2007 [37], the pore size found on the support would classify it as mesoporous because, for enzyme immobilization, the pore size needs to be between 2 and 50 nm. This porosity found can be correlated with the results presented in Table 5: when the immobilization of *Celluclast*® 1.5 L is at a concentration of 15 mg. mL$^{-1}$, the recovered activity of β-glucosidases is lower than that found for CMCase and FPase. Such a result may be related to the difference in the molecular weight of these enzymes and a possible limitation regarding the pore size of Accurel® MP1000, which can generate selective immobilization of a specific protein species to the detriment of another present in the extract. Proteins of small molecular weights tend to spread faster through the support and can lodge in the pores faster than those of higher molecular weights, thus being dominant at the beginning of the adsorption immobilization processes [40,41]. However, larger proteins may interact more intensely with the support due to their larger contact area and may be excluded from the pores and stick to the surface, and consequently, be more prone to deleterious actions of the microenvironment [40,42,43].

## 2.2. Protein Profile of the Enzyme Extracts ThI14-12 and Celluclast® 1.5 L

In order to understand the composition of both enzyme extracts in terms of protein composition and molecular weight, polyacrylamide gels (SDS-PAGE) were performed and compared with the molecular weight standards (Figure 5).

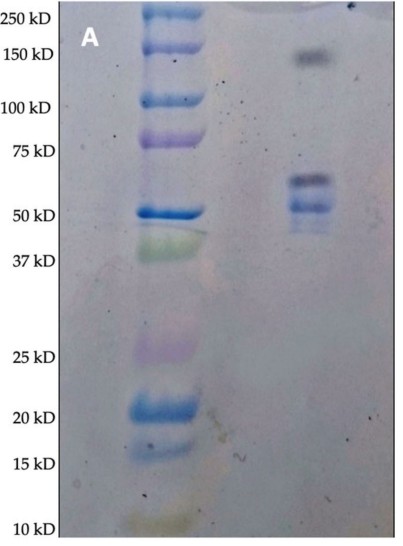
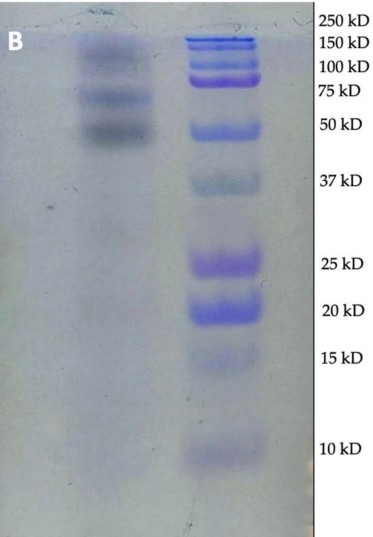

**Figure 5.** Polyacrylamide gels (SDS-PAGE). (**A**) An enzyme produced by the strain *Trichoderma harzianum* I14-12: the first lane shows the molecular mass pattern, while the second shows the enzymatic extract. (**B**) *Celluclast*® 1.5 L: the first lane shows the enzymatic extract, while the second one shows the molecular mass pattern.

As expected, both enzyme extracts are composed of a number of proteins. In both *Th* I-112 and *Celluclast*® 1.5 L extracts, three main bands are found, which differ in their molecular weights, according to the comparison with the applied standards. In the literature, the characterization of several fungal endoglucanases with sizes around 50 kD has been reported [44,45]. For *Trichoderma harzianum*, an enzyme with a molecular weight of 49.13 kD was found (PDB ID 4H7M), which may be associated with the band found near 50 kD [46,47]. Regarding exoglucanases, it has been reported that these enzymes have an average size of 46.28 kD, which can be associated with the weaker bands below 50 KD, as demonstrated in the extract Th-IL12 (Figure 5A) [48]. Importantly, there is also a second band between 50 and 75 kD, which may be associated with an isoform or the presence of a second exoglucanase in the extract. Regarding β-glucosidases originating from *Trichoderma harzianum*, proteins with molecular weights of 111.05 kD and 106.81 kD have been reported [49,50]. In the present extract, a strong band between 100 and 150 kD is noted and may be associated with this enzyme class.

Regarding the commercial extract *Celluclast*® 1.5 L, which presents enzymes produced by *Trichoderma reesei*, it has been reported to have a high activity of endoglucanase, which molecular weight is 68 kD and is most likely associated with the band present between 50 and 75 kD [51]. At 50 kD, a band is also evidenced, which probably indicates the presence of exoglucanase. For β-glucosidase, enzymes have also been reported for this species with molecular weights ranging between 100 and 150 kD, which corroborates with the bands found in this molecular weight range present in the gel for *Celluclast*® 1.5 L [52]. Therefore, it is likely that in the biocatalyst produced, there is a mixture of hydrophobic interactions, most pronounced for β-glucosidases, in addition to pore lodging for endo- and exoglucanases. Overall, the *Th* I14-12 extract shows lower molecular weights of its enzymes.

## 2.3. Immobilization of Enzymatic Extract ThI14-12 in Accurel® MP1000

Aiming to investigate the interference of these characteristics in the final biocatalyst, the extract *Th*I14-12 was also submitted to immobilization in Accurel® MP1000 under the same conditions determined for *Celluclast*® 1.5 L (Figure 6).

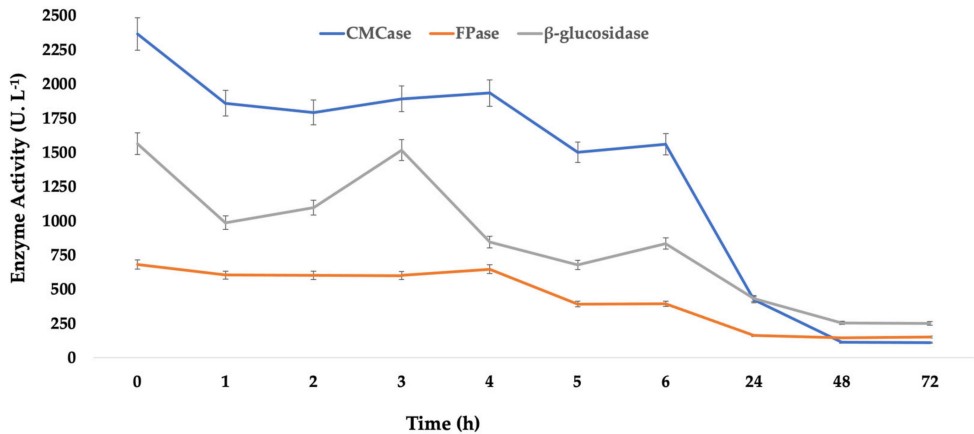

**Figure 6.** Kinetic profile of the enzymatic activity of *Th* I14-12 extract during immobilization in Accurel® MP1000. Immobilization conditions: 15 mg·mL$^{-1}$ of enzymatic extract in 50 mL of 50 mM citrate buffer at 40 °C and 100 rpm.

As it could be observed, for this enzyme extract, all the activities detected were reduced in the supernatant during the immobilization process by adsorption, where at 48 h, in the same way as occurred with the commercial preparation *Celluclast*® 1.5 L, it reached the equilibrium, expressing around 112 U.L$^{-1}$ of CMCase, 146 U.L$^{-1}$ of FPase, and 255 U.L$^{-1}$ of b-glucose. This last enzyme, unlike the others, showed a behavior of fluctuating activity during the immobilization kinetics, demonstrating that, in fact, this enzyme, by being larger than the others present in the extract, was probably adsorbed to the surface of the support, with this efficiency being more evident during the last hours of the process at the time during which the other enzymes were perhaps housed in the pores, thus making it possible for a better interaction of B-glucosidase with the surface of Accurel® MP1000.

Table 6 shows the results of immobilization efficiency, recovered activity, and support activity for the immobilized biocatalysts obtained in this work.

**Table 6.** Properties of the new biocatalysts produced in this work. Immobilization conditions: 15 mg·mL$^{-1}$ of each enzymatic extract in 50 mL of 50 mM citrate buffer at 40 °C and 100 rpm. *ACC-Celluclast*—biocatalyst obtained via the immobilization of *Celluclast*® 1.5 L in Accurel® MP1000; *ACC- Th* I14-12—biocatalyst obtained via the immobilization of the enzymatic extract *Th* I14-12 in Accurel® MP1000.

| Biocatalyst | Immobilization Yield (%) | Recovered Activity (%) | | | Support Activity (U.kg$^{-1}$) | | |
|---|---|---|---|---|---|---|---|
| | | CMCase | FPase | β-glucosidase | CMCase | FPase | β-glucosidase |
| **ACC-** *Celluclast* | 87 | 42.5 | 34.6 | 16.5 | 156.4 | 8.4 | 28.5 |
| **ACC-*Th* I14-12** | 95 | 54.6 | 48.6 | 20.4 | 254.6 | 10.4 | 60.6 |

As observed, both immobilized biocatalysts showed similar profiles of enzyme distribution on the support in the same concentration of applied protein, obtaining, at the end of the process, higher specific activities in CMCase, followed by β-glucosidase and FPase. However, the ACC-*Th* I14-12 biocatalyst presented a higher immobilization efficiency (95%)

and higher retention of activities, thus generating higher final activities on the support. Such data are pretty interesting since this extract was obtained using a wild strain and simple culture media, and thus, an immobilized biocatalyst was generated with the same enzymatic characteristics obtained in comparison with a commercial cellulolytic preparation, which could become a competitive option for various processes of hydrolysis of residual lignocellulosic biomass and its applications in bioprocesses. Moreover, the higher immobilization efficiencies found in this biocatalyst can possibly be a consequence of the difference between the molecular weights of the enzymes that compose it since the same concentration of proteins was applied in both immobilization processes. Thus, the enzymes present in the enzymatic extract *Th*I14-12 probably interacted more efficiently with the support, where more active conformations were generated for all enzymes, besides having been immobilized in a protein concentration that perhaps did not generate saturation of the support to the point of causing impediment or restriction of access of the enzymes to the substrate. These results are in line with several works reported in the literature for the immobilization of cellulases, where various immobilization techniques, such as covalent bonds, entrapment, and CLEA, are important to improve activity retention and enzyme activity [31,35,37,40–42]. However, this is the first work where complex extracts are immobilized using a hydrophobic adsorption methodology on Accurel® MP1000, a versatile support that is able to promote different hosting and adsorption interactions on its surface. However, most of the works demonstrate the results in global activity, thus not taking into account the isolated activity of the three main enzyme components of the extract, making this work important and unprecedented.

### 2.4. Characterization of the New Biocatalysts

In order to explore the catalytic properties of the new immobilized biocatalysts obtained in this work, thermal stability at temperatures ranging from 30 °C to 70 °C was evaluated in comparison with the free *Celluclast*® 1.5 L, which results are compiled in Table 7.

**Table 7.** Thermal stabilities of the new biocatalysts obtained in terms of residual activities. Immobilization conditions: 15 mg·mL$^{-1}$ of each enzymatic extract in 50 mL of 50 mM citrate buffer at 40 °C and 100 rpm. ACC-Celluclast—biocatalyst obtained via the immobilization of *Celluclast*® 1.5 L in Accurel® MP1000; ACC-Th I14-12—biocatalyst obtained via the immobilization of the enzymatic extract *Th*IL12 in Accurel® MP1000.

| Temperature (°C) | BIOCATALYST | | | | | | | | |
| | ACC-*Celluclast* | | | ACC-*Th*I14-12 | | | *Celluclast*® 1.5 L | | |
| | Residual Activity (%) | | | Residual Activity (%) | | | Residual Activity (%) | | |
| | CMCase | FPase | β-glucosidase | FPase | CMCase | β-glucosidase | CMCase | FPase | β-glucosidase |
| **30** | 78 | 81 | 81 | 85 | 90 | 85 | 67 | 78 | 86 |
| **40** | 95 | 92 | 100 | 100 | 98 | 95 | 100 | 100 | 100 |
| **50** | 94 | 91 | 98 | 97 | 95 | 96 | 88 | 90 | 91 |
| **60** | 85 | 88 | 95 | 96 | 95 | 89 | 56 | 57 | 54 |
| **65** | 76 | 85 | 86 | 88 | 90 | 79 | 35 | 38 | 40 |
| **70** | 12.5 | 4.5 | 19 | 14 | 12 | 7.5 | 5 | 2 | 3 |

As a result, it could be observed that the immobilized enzymes showed greater thermal stabilities at higher temperatures when compared to the commercial *Celluclast*® 1.5 L. At 40 °C, the free commercial extract presented the maximum catalytic activities of endoglucanases and exoglucanases, besides β-glucosidase which suffered decreases with an increase in temperature, losing more than 50% of enzymatic activities at 60 °C and more than 90% at 70 °C. Comparing these data with the ACC-*Th*I14-12 biocatalyst, immobilization promoted the maintenance of the catalytic activities of the enzymes at temperatures up to 60 °C, where more than 95% of the activities of endoglucanases and

exoglucanases, as well as β -glucosidase, were still detected under the conditions studied. At 65 °C, 88% of FPase, 90% of CMCase, and 79% of β-glucosidase activities were still found, demonstrating relative thermal stability at this temperature and showing its high potential for application in bioprocesses requiring this temperature. At 70 °C, the enzyme that suffered the greatest decrease in relative activity was β-glucosidase, probably due to the fact that it is supposedly spread over the surface of the support, thus being more exposed to the deleterious actions of the environment. Similar profiles of thermal stability and maintenance of residual activity were obtained for the ACC-Celluclast biocatalyst, demonstrating that the immobilized home-made extract presented competitive characteristics when compared to a commercial preparation, with high retention of activity, thermal stability, and efficiency of immobilization, thus being a potential candidate as a biocatalyst in new bioprocesses.

Operational stability and reusability potential are crucial factors to reduce long-term costs in processes involving immobilized enzymes. In this way, aiming to glimpse the potential application for the new biocatalyst ACC-*Th*I14-12, it was submitted to the standard test of CMC hydrolysis for eight hours and 60 °C temperature, when the biocatalyst still kept a high recovered activity in the support. The results, expressed as glucose yield released at each reaction cycle, are shown in Figure 7.

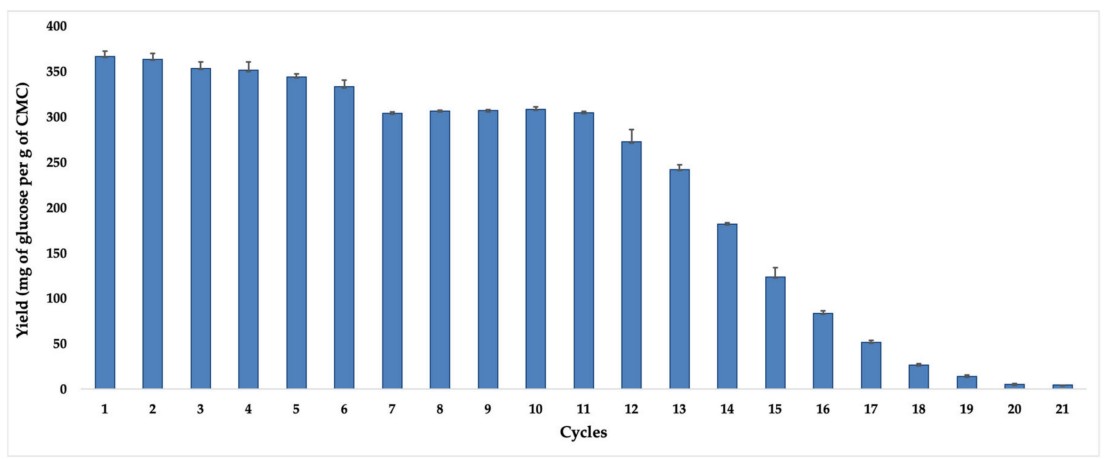

**Figure 7.** Recycles and operational stability of the biocatalyst ACC-*Th*I14-12 based on CMC hydrolysis at a pH of 4, for 8 h, and at 60 °C.

As observed, in the first hydrolysis cycle, a maximum glucose concentration of 366 mg per g of CMC was obtained, which remained without significant reduction in efficiencies until the sixth cycle. Until the 11th recycling, the ACC-*Th*I14-12 biocatalyst was still able to generate glucose concentrations of around 300 mg per g of CMC. From the 12th recycle onward, significant losses in hydrolysis efficiency were detected, probably due to denaturation of the immobilized enzymes or desorption and leaching effects since the methodology applied was surface adsorption [53]. Such results are superior to the work by Mo et al. [54], who immobilized cellulases in a preparation of chitosan/magnetic porous biochar as a support. As for the recycle results, a maximum of 330.9 mg of glucose per g of CMC was obtained in the reaction cycles at 24 h and 40 °C, where the biocatalyst, obtained by covalent bonding, was able to maintain its operational stability for ten cycles. Therefore, our biocatalyst is competitive and presents operational stability for hydrolytic processes.

### 3. Materials and Methods

#### 3.1. Materials

*Celluclast*® 1.5 L and all the reagents used in the experiments for obtaining the cellulolytic extract were purchased from Sigma-Aldrich (Rio de Janeiro, Brazil). Precision Plus Protein™ Kaleidoscope Standards for SDS-PAGE assays were purchased from Bio-Rad Lab-

oratories. Accurel® MP 1000 was purchased from Membrane GmbH (Germany). Sugarcane straw was obtained from the National Laboratory of Biorenewables.

### 3.2. Production of Cellulolytic Enzymatic Extract from Trichoderma harzianum I14-12

The enzymatic cocktail from *Trichoderma harzianum* I14-12 used in the experiments was obtained from the submerged fermentation of the wild strain collected from the soil of the Itatiaia National Park (Brazil) using steep corn liquor (acquired from Sigma-Aldrich Brazil) and sugarcane straw (obtained from the National Laboratory of Biorenewables). For this purpose, the fermentations were performed in 125 mL Erlenmeyer flasks, starting with an inoculation of 25 µL of spore suspension in 25 mL of Mandels' medium, plus 2% ($w/v$) of sugarcane straw as a carbon source and 1% ($w/v$) of millet as a nitrogen source. The system was incubated in a chilled shaker at 28 °C and 200 rpm for 72 h. Al shares of 650 µL were removed and centrifuged (10,000 rpm for seven minutes) to obtain the supernatants, which were submitted to the enzymatic activity detection assays described in the following sections.

### 3.3. Immobilization Procedures

For initial screening, immobilization of the cocktail *Celluclast*® 1.5 L via hydrophobic adsorption was carried out in 250 mL Erlenmeyer flasks containing 2 g of Accurel® MP1000 and the appropriate amount of enzymatic extract diluted in 25 mM and 50 mM citrate buffer for an initial concentration of 45mg·mL$^{-1}$ to obtain a final volume of 50 mL at 20 °C or 40 °C under a shaker at 100 rpm. Aliquots of the supernatant were taken in order to quantify the immobilization efficiency through the concentration of proteins and enzyme residual activity. At the end of the immobilization processes, the resulting biocatalysts were filtered under vacuum and subsequently washed three times with 50 mL of appropriate buffer and dried at room temperature, being subsequently investigated in terms of CMCase, FPase, and β-glucosidase activities. For all subsequent immobilization experiments, the optimal conditions were applied with an appropriate amount of proteins and an appropriate proportion of the enzyme to the support.

### 3.4. Quantification of Protein Concentration

The quantification of the concentration of proteins from the cocktail of Celluclast® 1.5 L and the supernatants during the immobilization process was carried out according to the method of Bradford [55], where the protein content was estimated based on the average of the calibration curve obtained using bovine albumin serum (BSA) as a standard at 595 nm (Micronal B-520).

### 3.5. Enzymatic Assays

#### 3.5.1. CMCase

CMCase activity was determined according to the protocol described by Ghose [56], which is a modification of the dinitrosalicylic acid (DNS) method [57].

#### 3.5.2. FPase

FPase activity was determined using the filter paper method: the reaction mixture contained filter paper Whatman N°1 (1.0 cm × 1.0 cm) as a substrate in 40 µL of 50 mM sodium citrate buffer (pH of 4.8) and 20 µL of the enzymatic extract. The system was incubated for 60 min at 50 °C [58], followed by the determination of glucose content using the DNS method [57].

#### 3.5.3. β-glucosidase

β-glucosidase activity was determined via hydrolysis using the *p*-nitrophenyl-β-D-glucopyranoside method according to the protocol of Silva [59]. The system was prepared with 650 µL of distilled water, 200 µL of 0.5 M sodium acetate buffer (pH 5), and 50 µL of the enzymatic extract and left to rest for five min. at 50 °C for complete temperature stabi-

lization. Then, 100 μL of a 0.5 mg·mL$^{-1}$ solution of *p*-nitrophenyl-β-D-glucopyranoside in 0.5 M sodium acetate buffer was added. The system was incubated for 10 min. The reaction was interrupted with the addition of 500 μL of a Na$_2$CO$_3$ 1 M solution (pH of 10). The quantification of released *p*-nitrophenyl during the process was performed at 420 nm in a spectrophotometer.

*3.6. Immobilization Yield and Recovery Activity on the Support*

The immobilization yield (R) was defined as the amount of theoretically immobilized enzyme [29], calculated using Equation (1) as follows:

$$R(\%) = \left( 1 - \frac{NAds}{Ads} \right) * 100 \tag{1}$$

where *NAds* is the amount of protein (mg/mL) in the supernatant after a certain period of immobilization, while *Ads* is the initial amount of protein (mg·mL$^{-1}$).

The recovered activities of the support were defined as the percentage of the immobilized enzyme, taking into account the initial activity of the free enzyme. It is calculated using Equation (2) as follows:

$$Rec\ (\%) = \frac{A_{imob}}{A_i} * 100 \tag{2}$$

where *Ai* is the activity (U.L$^{-1}$) of the initial enzyme, and *Aimob* is the activity of the immobilized enzyme (U.kg$^{-1}$).

*3.7. Support Analysis*

3.7.1. Scanning Electron Microscopy (SEM)

The Accurel® MP 1000 support, before and after enzyme immobilization, was submitted for visualization of its pores. The equipment used was the Hitachi SEM TM3030Plus, Tokyo, Japan, and before analysis, the samples were metalized using the Quorum model Q 150R ES metallizer.

3.7.2. Thermogravimetric Analysis (TG)

TG analysis of Accurel MP® 1000 was carried out to analyze mass variation up to a temperature of 600 °C. This procedure was performed using the Shimadzu Thermogravimetric Analyzer TGA-50 equipment, Kyoto, Japan.

3.7.3. Textural Characterization (BET)

The textural characterization of Accurel® MP 1000 was determined based on porosimetry using a Quantachrome® porosimeter (NOVA-1200), which aimed to evaluate the capacity of the material for adsorption and subsequent desorption of nitrogen at 77 K, thereby forming isotherms of adsorption and desorption. The degassing of the sample under vacuum was performed at a constant temperature of 150 °C for two hours to eliminate possible contaminants. From the adsorption isotherm data, the surface area, total volume, and average pore diameter of the material could be determined according to the method devised by Brunauer, Emmet, and Teller (BET) [60].

3.7.4. Determination of Hydrophobicity of the Support

The relative hydrophobicity of Accurel® MP 1000 was determined based on the adsorption of Rose Bengal dye according to the procedure described by Lima et al., 2015 [61]: 0.150 g of Accurel® MP 1000 was added into Erlenmeyer flasks of 125 mL containing 20 mL of the dye solution at 20 μg·mL$^{-1}$. The flasks were kept under stirring for 1 h at room temperature. The samples were filtered, and the supernatant was used to quantify the concentration of the dye based on the difference between the initial and final absorbances at 549 nm and by comparing them with the respective standard curve of the dye. The

adsorption efficiency was calculated as the amount of Rose Bengal dye adsorbed per unit area of the support.

### 3.8. SDS-PAGE Analysis

In order to perform a comparative analysis of the protein profile of the enzymatic cocktail *Celluclast*® 1.5 L and the enzymatic extract obtained from the strain *Trichoderma harzianum* I14-12, a polyacrylamide gel electrophoresis (SDS-PAGE) was performed. The samples were prepared with a sample buffer prepared according to the Bio-Rad Bulletin number 6040 of biorad and boiled for 5 min before being applied to the gel, which was also made according to the Bio-Rad protocol. The electrophoretic run took place at 150 V and 50 mA current in a vertical bowl system (Mini-PROTEAN Tetra Cell, Bio-Rad). The gel was stained with a stain solution for 30 min under agitation and discolored with a decolorizing solution, also with agitation, until the appearance of bands. Both solutions were prepared according to the Department of Chemistry of the University of Florida. Precision Plus Protein™ Kaleidoscope Standards (Biorad, Hercules, CA, USA) was used as a molecular mass marker, covering the 10–250 kDa range.

### 3.9. Recycles of Biocatalysts

Aiming to investigate the reusability of the ACC-ThI14-12 biocatalyst, successive hydrolysis of CMC was performed according to the modification of the protocol developed by Zang et al., 2014 [53]. For this purpose, 5 g of the ACC-ThI14-12 biocatalyst was added and mixed with 10 mL of 1% ($m/v$) CMC solution (pH of 4) for 8 h at 60 °C. Then, the immobilized cellulase was filtered under a vacuum and washed separately for further recycles. Its recyclability was evaluated based on the production of reducing sugars from each cycle [57].

## 4. Conclusions

In the present work, the cellulolytic enzyme extract obtained from the wild-type *Trichoderma harzianum* strain I14-12 showed much lower protein content than the commercial enzyme preparation *Celluclast*® 1.5 L, but it had similar composition and competitive activities of exoglucanases, endoglucanases, and β-glucosidases.

During the immobilization of the enzymatic extracts in Accurel® MP 1000, the concentration of 15 mg·mL$^{-1}$ of proteins and high ionic strengths generated biocatalysts with high activity retention, thermal stability, and immobilization efficiency, where the size and molecular weight of the different enzymes present in the extracts were responsible for different hydrophobic interactions and lodging of the proteins in the support; the ACC-*Th*I14-12 biocatalyst presented better thermal stabilities and recovered activities, besides exhibiting operational stability for more than ten reaction cycles.

Therefore, this work reports for the first time the immobilization of cellulolytic enzymes on Accurel® MP 1000 and the obtainment of a biocatalyst competitive with commercially available enzymes, which may provide a lower cost option in the reuse of waste biomass.

**Supplementary Materials:** The following supporting information can be downloaded at https://www.mdpi.com/article/10.3390/reactions4020019/s1. Figure S1. Total protein concentration shown by *Celluclast*® 1.5 L cocktail immobilized in Accurel® MP 1000 across time in each studied condition. Figure S2. Residual enzyme activity in the supernatant during the immobilization of *Celluclast*® 1.5 L cocktail in Accurel MP1000. Immobilization conditions: 45 mg·mL$^{-1}$ of enzymatic extract in 50 mL of 50 mM citrate buffer at 40 °C and 100 rpm.

**Author Contributions:** Conceptualization: R.P.d.N. and I.I.J.; methodology: J.R.S.B., A.C., J.C.S.S., B.D.R. and I.I.J.; validation: J.R.S.B., A.C., J.C.S.S. and B.D.R.; formal analysis: J.R.S.B., A.C. and J.C.S.S.; investigation: J.R.S.B., A.C., J.C.S.S., B.D.R., R.P.d.N. and I.I.J.; resources: B.D.R., R.P.d.N. and I.I.J.; data curation: J.R.S.B. and I.I.J.; writing—original draft preparation: J.R.S.B. and I.I.J.; writing—review and editing: I.I.J.; supervision: R.P.d.N. and I.I.J.; project administration: R.P.d.N. and I.I.J.; funding acquisition: R.P.d.N. and I.I.J. All authors have read and agreed to the published version of the manuscript.

**Funding:** This study was funded by the Brazilian government through the Fundação Carlos Chagas Filho de Amparo à Pesquisa do Estado do Rio de Janeiro—Grant Numbers: E-26/203.267/2017 and E-26/201.367/2022; the Conselho Nacional de Desenvolvimento Científico e Tecnológico CNPq Brazil—Grant Number 429974/2018-3; and the Coordenação de Aperfeiçoamento de Pessoal de Nível Superior.

**Conflicts of Interest:** The authors declare no conflict of interest.

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
