# Peer review of "Immobilization of Cellulolytic Enzymes in Accurel® MP1000"

_reactions, doi:10.3390/reactions4020019_

Round 1
Reviewer 1 Report
Cellulases are important bio-enzyme for industry applications. Immobilization is a useful method to consider both the catalysis efficiency and stability. Authors immobilized cellulases on Accurel® MP1000, obtaining good catalysis efficiency and enzyme stability, which is valuable for cellulases application in biomass degradation and conversion.
Author Response
Dear Referee,
Thank you very much for your comment and for reviewing our manuscript. We appreciate your work and availability.
Reviewer 2 Report
An experimental article entitled "Immobilization of Cellulolytic Enzymes in Accurel® MP1000" corresponds to "Reactions" because the authors present the results of immobilization of cellulolytic enzymes of the commercial enzyme cocktail Celluclast® 1.5 L in comparison to the cellulolytic enzyme cocktail produced from the wild strain Trichoderma harzianum I14-12 in Accurel® MP1000. The authors found that the cellulolytic enzymatic extract of the wild strain Trichoderma harzianum I14-12 proved to be a promising biocatalyst for bioprocesses of hydrolysis of lignocellulosic biomass, despite a much lower protein content than the commercial enzyme preparation.
Below are some remarks:
⦁ The introduction describes the industrial significance of cellulases, their characteristics and substantiates the need for immobilization of cellulases. However, examples of the immobilization of these enzymes on other carriers and the results achieved in this area are not described. It is necessary to briefly touch on this issue. You also need to justify the choice of Accurel® MP1000 media.
⦁ The introduction states that the enzyme extract was obtained using sugarcane straw and millet as carbon sources. materials and methods indicate that it is obtained using corn extract and sugarcane straw. The inconsistency needs to be corrected.
⦁ The authors investigate the immobilization of the cellulolytic enzymatic extract of the wild strain Trichoderma harzianum I14-12 and show its effectiveness. In materials and methods, the method of its preparation is briefly described. In the discussion of the results, it is written that the drug was obtained under simple fermentation conditions ... I would like a more detailed description of the method for obtaining the studied cellulolytic cocktail or provide a corresponding link.
⦁ The conclusions indicated that the enzymatic extract of the wild strain Trichoderma harzianum I14-12 proved to be a promising biocatalyst for bioprocesses of hydrolysis of lignocellulosic biomass. However, there are no examples of hydrolysis of specific types of lignocellulose in the work. It would be interesting if the authors presented the results of enzymatic hydrolysis of real lignocellulose with a new enzyme cocktail.
Author Response
An experimental article entitled "Immobilization of Cellulolytic Enzymes in Accurel® MP1000" corresponds to "Reactions" because the authors present the results of immobilization of cellulolytic enzymes of the commercial enzyme cocktail Celluclast® 1.5 L in comparison to the cellulolytic enzyme cocktail produced from the wild strain Trichoderma harzianum I14-12 in Accurel® MP1000. The authors found that the cellulolytic enzymatic extract of the wild strain Trichoderma harzianum I14-12 proved to be a promising biocatalyst for bioprocesses of hydrolysis of lignocellulosic biomass, despite a much lower protein content than the commercial enzyme preparation.
Below are some remarks:
⦁ The introduction describes the industrial significance of cellulases, their characteristics and substantiates the need for immobilization of cellulases. However, examples of the immobilization of these enzymes on other carriers and the results achieved in this area are not described. It is necessary to briefly touch on this issue. You also need to justify the choice of Accurel® MP1000 media.
Answer: Dear referee, thank you for your questioning.
Our group decided to apply the Accurel MP1000 support, since it is a highly porous, hydrophobic and relatively inexpensive material, which our research group has experience in immobilizing lipases and transaminases, with promising results. No work has explored the immobilization of cellulases on this support so far.
The most current references focus on immobilization of cellulases by covalent bonds or other methodologies different from adsorption, and apply relatively purified cellulases, and not compound enzyme extracts, like ours. For this reason, we decided to study the immobilization process of Celluclast 1.5L, in addition to our cellulolytic extract obtained by the wild-type Trichoderma harzianum strain I14-12 on this support, in order to understand the consequences in terms of activity retention and operational stability. We have rewritten the last paragraph of the introduction further justifying the application of Accurel MP 1000.
⦁ The introduction states that the enzyme extract was obtained using sugarcane straw and millet as carbon sources. materials and methods indicate that it is obtained using corn extract and sugarcane straw. The inconsistency needs to be corrected.
Answer: Dear referee, indeed, there was a mistake in the introduction and materials and methods sections. We corrected these informations.
⦁ The authors investigate the immobilization of the cellulolytic enzymatic extract of the wild strain Trichoderma harzianum I14-12 and show its effectiveness. In materials and methods, the method of its preparation is briefly described. In the discussion of the results, it is written that the drug was obtained under simple fermentation conditions ... I would like a more detailed description of the method for obtaining the studied cellulolytic cocktail or provide a corresponding link.
Answer: Dear referee, thank you for your observation, in fact, to obtain the enzymatic extract, sugar cane straw was used as carbon source and millet as nitrogen source. This data has been added in the material and methods section of the manuscript for better understanding.
⦁ The conclusions indicated that the enzymatic extract of the wild strain Trichoderma harzianum I14-12 proved to be a promising biocatalyst for bioprocesses of hydrolysis of lignocellulosic biomass. However, there are no examples of hydrolysis of specific types of lignocellulose in the work. It would be interesting if the authors presented the results of enzymatic hydrolysis of real lignocellulose with a new enzyme cocktail.
Answer: Answer: Dear referees, thank you for the observation and questioning. In fact, in this manuscript, we did not have enough time to present the results of hydrolysis of real lignocellulose with the new biocatalysts obtained both free and immobilized, mainly due to the fact that the time for the submission of corrections and suggestions by the referees is relatively short for the realization of a large batch of experiments. However, we know that the free enzyme Celluclast 1.5L cocktail has shown promising results when compared to the commercial enzyme used as a standard of comparison in this work. As for the obtained immobilized biocatalyst, we have studied its action in biomass after treatment with eutectic solvents as initial pre-treatment, and this way, the biocatalyst is applied in liquid phase. Aiming at the study of operational stability and pointing out the promising potential of our biocatalysts, we proposed in this paper the hydrolysis of CMC, which gave us indications of being a robust biocatalyst. However, we are still finalizing this step of the work.
Reviewer 3 Report
Manuscript of Baruque et al. is devoted to an important aspect of modern biotechnology - the immobilization of enzymes and corresponds to the profile of the journal Reactions.
Key notes:
- Based on the logic of the authors who claim that the use of ACC-ThI1412 biocatalyst in the conversion of cellulose materials is promising, why is there no data on the use of the obtained immobilized enzymes for specific industrial substrates? Taking into account the heterogeneity of the processes of hydrolysis of lignocellulosic materials, as well as the presence of CBM in some cellulases, the use of immobilized preparations may not be as effective as the activity values show.
- The use of Celluclast as a comparison does not seem reasonable. There are now more modern cocktails on the market, such as the Cellic CTec series and others. In addition, it is also derived from filamentous fungi of the genus Trichoderma.
Section notes:
Abstract: One proposal mentions FPAse along with CMCase and b-glucosidase. FPAse is known to be the result of a variety of enzymatic reactions catalyzed by various cellulases, including endoglucanases (which exhibit CMC activity) and b-glucosidase. It is more correct to simply talk about activities on certain substrates.
Introduction: There is no mention of hemicellulases, in particular xylanases. Most likely, they are present in the preparations under study, and their role is no less important in the conversion of lignocellulosic raw materials.
Results: It is not entirely clear what the authors wanted to show using SEM analysis? Just the surface of the immobilizing agent? It is difficult to assess the effectiveness of immobilization in the presented images. These images can be used simply as an additional feature of the Accurel® MP 1000.
It is required to carefully check the location of the tables in the text of the manuscript, as well as their captions and column names (Table 7 is used 2 times by CMCase for one sample).
Conclusion The first sentence refers to the prospects of a new drug based on Trichoderma, but this is not the main goal of the article, especially since there are no results on complex real lignocellulosic substrates. The article is devoted to immobilization!
Author Response
Answers to referee 3
Manuscript of Baruque et al. is devoted to an important aspect of modern biotechnology - the immobilization of enzymes and corresponds to the profile of the journal Reactions.
Key notes:
- Based on the logic of the authors who claim that the use of ACC-ThI1412 biocatalyst in the conversion of cellulose materials is promising, why is there no data on the use of the obtained immobilized enzymes for specific industrial substrates? Taking into account the heterogeneity of the processes of hydrolysis of lignocellulosic materials, as well as the presence of CBM in some cellulases, the use of immobilized preparations may not be as effective as the activity values show.
Answer: Dear referee, Thank you for the note and the discussion. Indeed, our group has not demonstrated in this paper any direct application results on crude lignocellulosic biomass or substrates more complex than carboxymethylcellulose, evidenced in our manuscript. We are working on this application as part of a second paper, where lignocellulosic biomass is being dissolved in eutectic solvents, and the application of our biocatalyst has been quite important in this regard. In addition, the short turnaround time for the revision of this manuscript also limits the addition of more complex experimental data such as this. Thus, we focused our attention on studying the effects of the immobilization process of the cellulolytic enzyme extracts on the Accurel® MP1000 support, which had never been used before for this group of enzymes, understanding the inherent effects of the process on enhancing and retaining the catalytic activities present, and performing the recycle study of the final biocatalyst, in addition to its characterization, the work being a proof of concept.
- The use of Celluclast as a comparison does not seem reasonable. There are now more modern cocktails on the market, such as the Cellic CTec series and others. In addition, it is also derived from filamentous fungi of the genus Trichoderma.
Answer: Dear referee, thank you for your question. Indeed, there are more modern enzyme cocktails on the market derived from fungi of the genus Trichoderma. However, our laboratory did not have these at the time of the preparation of this manuscript, only the Celluclast® 1.5 L cocktail, which also has a relevant industrial application, besides being a comparative standard for the enzymatic hydrolysis of complex cellulosic materials, given its composition and high enzymatic activity, as determined in our manuscript. As described above, we also did not have time to perform a more intensified comparison with other preparations, and in future application steps, we will perform comparison with the Cellic® CTec2 cocktail, which is closest to our research scenario. However, it is important to note that the cost of these enzymes is relatively high, and impacts the final process. Our work proposes to obtain an enzyme preparation rich in cellulolytic activities, obtained from a wild strain of the genus Trichoderma, and subsequent immobilization by hydrophobic adsorption on support Accurel® MP1000, with promising results. Therefore, our work aims at a proof of concept about the potential of our enzyme extract obtained through submerged fermentation using residual substrates.
Section notes:
Abstract: One proposal mentions FPAse along with CMCase and b-glucosidase. FPAse is known to be the result of a variety of enzymatic reactions catalyzed by various cellulases, including endoglucanases (which exhibit CMC activity) and b-glucosidase. It is more correct to simply talk about activities on certain substrates.
Answer: Dear referee, thank you for your comment. We have restructured the abstract section to better portray this information.
Introduction: There is no mention of hemicellulases, in particular xylanases. Most likely, they are present in the preparations under study, and their role is no less important in the conversion of lignocellulosic raw materials.
Answer: Dear referee, thank you for pointing this out. Certainly, hemicellulases, like xylanases play a crucial role in the complete hydrolysis of raw or residual lignocellulosic materials. However, since the focus of our study was on the detection of cellulase activities in our enzyme extracts and subsequent immobilization, we restricted the introduction of this manuscript to these enzymes. For the sake of completeness, we have also added hemicellulases in this section of the paper.
Results: It is not entirely clear what the authors wanted to show using SEM analysis? Just the surface of the immobilizing agent? It is difficult to assess the effectiveness of immobilization in the presented images. These images can be used simply as an additional feature of the Accurel® MP 1000.
Answer: Dear referee, thank you for the discussion. In fact, the scanning electron microscopy applied in this work was merely an accessory technique to observe the shape of the Accurel MP1000 support before and after the immobilization process, in order to understand that this is an irregularly shaped and highly porous support, whose porosity is highly distributed along the support. In the higher magnifications it was also possible to observe some filled pores, which was probably due to the immobilization process of the enzyme extract, but in no way, this technique was determinant to affirm that the immobilization process happened.
It is required to carefully check the location of the tables in the text of the manuscript, as well as their captions and column names (Table 7 is used 2 times by CMCase for one sample).
Answer: Dear Referee, thank you for the note. We have renumbered the tables correctly in the manuscript, as well as adjusted the citations throughout the text and corrected the table 7. In fact, there were some erroneous citations.
Conclusion The first sentence refers to the prospects of a new drug based on Trichoderma, but this is not the main goal of the article, especially since there are no results on complex real lignocellulosic substrates. The article is devoted to immobilization!
Answer: Dear referee, thank you for your question. We have modified the first paragraph of the conclusion to better align it with the purposes of the article.
Round 2
Reviewer 2 Report
The authors took seriously the comments made and made changes to the text that answer the questions posed.
Reviewer 3 Report
Manuscript of Baruque et al. has been substantially revised in accordance with the comments. In cases where changes were not made, the authors gave reasoned answers. The manuscript is recommended for publication in Reactions.